# Revisiting the Temporal Modeling in Spatio-Temporal Predictive Learning under a Unified View

## Abstract

Spatio-temporal predictive learning plays a crucial role in self-supervised learning, with wide-ranging applications across a diverse range of fields. Previous approaches for temporal modeling fall into two categories: recurrent-based and recurrent-free methods. The former, while meticulously processing frames one by one, neglect short-term spatio-temporal information redundancies, leading to inefficiencies. The latter naively stack frames sequentially, overlooking the inherent temporal dependencies. In this paper, we re-examine the two dominant temporal modeling approaches within the realm of spatio-temporal predictive learning, offering a unified perspective. Building upon this analysis, we introduce USTEP (Unified Spatio-TEmporal Predictive learning), an innovative framework that reconciles the recurrent-based and recurrent-free methods by integrating both micro-temporal and macro-temporal scales. Extensive experiments on a wide range of spatio-temporal predictive learning demonstrate that USTEP achieves significant improvements over existing temporal modeling approaches, thereby establishing it as a robust solution for a wide range of spatio-temporal applications.

## 1 Introduction

In an era where data is continually streaming in, there is an increasing demand to not only understand the present but to also predict the future. By leveraging historical video data, spatio-temporal predictive learning strives to forecast subsequent sequences in an unsupervised manner Finn et al. (2016); Hsieh et al. (2018); Locatello et al. (2019); Greff et al. (2019); Mathieu et al. (2019); Khemakhem et al. (2020); Castrejon et al. (2019). With real-world applications extending from forecasting weather patterns Shi et al. (2015); Gao et al. (2022b); Rasp et al. (2020) to predicting traffic flows Fang et al. (2019); Wang et al. (2019b) and simulating physical interactions Lerer et al. (2016); Finn et al. (2016), the ramifications of advancements in this domain are profound.

The path to achieving accurate spatio-temporal predictions has been fraught with challenges. Traditional approaches have typically oscillated between two primary temporal modeling methodologies: recurrent-based and recurrent-free methods. The recurrent-based methods Shi et al. (2015); Wang et al. (2017); Lotter et al. (2017); Wang et al. (2018a; 2021; 2019a; 2018b); Jin et al. (2020); Babaeizadeh et al. (2021) meticulously process frames one by one, ensuring that temporal relationships across each timestep are captured. Yet, they often grapple with inefficiencies arising from the redundant short-term spatio-temporal information and challenges in preserving global information from preceding time steps. Conversely, the recurrent-free methods Tan et al. (2023a); Gao et al. (2022a); Tan et al. (2022; 2023b), while alleviating the inefficiencies of their recurrent counterparts, fall short in capturing the inherent temporal dependencies. By stacking frames in a naive manner, these models may overlook the intricate dance of cause and effect played out over time.

In this work, we revisit the foundational principles of temporal modeling in spatio-temporal predictive learning, dissecting the merits and demerits of the prevailing approaches. We introduce the concept of a temporal segment, defined as a subsequence encompassing a series of continuous frames. To refine our understanding further, we formally identify and delineate two temporal scales: the micro-temporal scale, which focuses on immediate, sequential dependencies, and the macro-temporal scale, which encapsulates long-range global patterns.

Our analysis reveals that recurrent-based methods primarily concentrate on micro-temporal scales, adept at capturing instantaneous interactions but often lacking in long-term insight. On the other hand, recurrent-free methods seem to excel in considering macro-temporal scales, but their neglect of immediate temporal dependencies often leads to a loss in the richness of the predicted sequences. This discrepancy between the two paradigms presents an evident gap in current methodologies.

To bridge this gap, we introduce USTEP (Unified Spatio-TEmporal Predictive learning), a novel framework that takes into account both micro- and macro-temporal scales. By doing so, USTEP achieves a balanced trade-off between predictive performance and computational efficiency. The architecture of USTEP is designed to integrate seamlessly the strengths of both recurrent-based and recurrent-free methods, while also introducing new mechanisms that enhance the model's ability to generalize across various spatio-temporal scales. We conduct a diverse range of spatio-temporal predictive tasks and the experimental results demonstrate its superior performance over existing methods, not just in terms of accuracy but also in computational efficiency. We find that USTEP can achieve state-of-the-art performance with moderate computational resources, thereby establishing it as a potent solution for practical, real-world applications.

## 2 RELATED WORK

### 2.1 RECURRENT-BASED SPATIO-TEMPORAL PREDICTIVE LEARNING

Recurrent-based models have made significant strides in the field of spatio-temporal predictive learning. Drawn inspiration from recurrent neural networks Hochreiter & Schmidhuber (1997), VideoModeling Marc'Aurelio Ranzato et al. (2014) incorporates language modeling techniques and employs quantization of picture patches into a comprehensive dictionary for recurrent units. ConvLSTM Shi et al. (2015) leverages convolutional neural networks to model the LSTM architecture. PredNet Lotter et al. (2017) persistently predicts future video frames using deep recurrent convolutional neural networks with bottom-up and top-down connections. PredRNN Wang et al. (2017) proposes a Spatio-temporal LSTM (ST-LSTM) unit that extracts and memorizes spatial and temporal representations simultaneously, and its following work PredRNN++ Wang et al. (2018a) further introduces gradient highway unit and Casual LSTM to capture temporal dependencies adaptively. E3D-LSTM Wang et al. (2018b) designs eidetic memory transition in recurrent convolutional units. PredRNN-v2 Wang et al. (2021) has expanded upon PredRNN by incorporating a memory decoupling loss and a curriculum learning technique. However, recurrent-based models struggle with capturing long-term dependencies. Moreover, they tend to be computationally intensive, especially when scaled to high-dimensional data, thereby limiting their practical applicability.

### 2.2 RECURRENT-FREE SPATIO-TEMPORAL PREDICTIVE LEARNING

Instead of employing computationally intensive recurrent methods for spatio-temporal predictive learning, alternative approaches such as PredCNN Xu et al. (2018) and TrajectoryCNN Liu et al. (2020) utilize convolutional neural networks for temporal modeling. SimVP Gao et al. (2022a); Tan et al. (2022) represents a seminal work that incorporates blocks of Inception modules within a UNet architecture. Additionally, TAU Tan et al. (2023a) introduces a temporal attention unit designed for efficient spatio-temporal modeling. However, these recurrent-free models have limitations in capturing fine-grained temporal dependencies for naive temporal modeling. Furthermore, they lack flexibility in decoding, as they are designed with output lengths that match their input lengths.

### 2.3 EFFICIENT RECURRENT NEURAL NETWORK

Recently, RWKV Peng et al. (2023) and RetNet Sun et al. (2023) revisit RNNs and propose RNN architectures that can achieve performance in sequence modeling comparable to Transformers Vaswani et al. (2017). Mega Ma et al. (2022) proposes a chunk-wise recurrent design, using the moving average equipped gated attention mechanics to capture long-range dependencies in sequential data across various modalities. While these prior works demonstrate that well-designed recurrent architectures can be both effective and efficient, USTEP goes a step further by synergizing recurrent and recurrent-free paradigms. This hybrid approach allows USTEP to capture both micro- and macro-temporal scales, offering a nuanced and robust framework for spatio-temporal prediction.

## 3 BACKGROUND

We formally define the spatio-temporal predictive learning problem, inspired by existing works Gao et al. (2022a); Tan et al. (2023b). Consider an observed sequence of frames $\mathcal{X}^{t,T} = \{\boldsymbol{x}^i\}_{t-T+1}^t$ at a specific time $t$, comprising the past $T$ frames. Our objective is to forecast the subsequent $T'$ frames, denoted as $\mathcal{Y}^{t+1,T'} = \{\boldsymbol{x}^i\}_{t+1}^{t+T'}$. Each frame $\boldsymbol{x}_i$ is generally an image in $\mathbb{R}^{C\times H\times W}$, with $C$ being the number of channels, $H$ the height, and $W$ the width. In the tensorial representation, the observed and predicted sequences are represented as $\mathcal{X}^{t,T} \in \mathbb{R}^{T\times C\times H\times W}$ and $\mathcal{Y}^{t+1,T'} \in \mathbb{R}^{T'\times C\times H\times W}$.

Given a model with learnable parameters $\Theta$, we seek to find a mapping $\mathcal{F}_\Theta : \mathcal{X}^{t,T} \mapsto \mathcal{Y}^{t+1,T'}$. This mapping is realized through a neural network model that captures both spatial and temporal dependencies within the data. The model is trained to minimize a loss function $\mathcal{L}$ that quantifies the discrepancy between the predicted and ground-truth future frames, formulated as:

$$\min_\Theta \mathcal{L}(\mathcal{F}_\Theta(\mathcal{X}^{t,T}), \mathcal{Y}^{t+1,T'}), \tag{1}$$

where $\mathcal{L}$ is chosen to evaluate the quality of the predictions in both spatial and temporal dimensions.

**Definition 3.1** (temporal segment). *A temporal segment is defined as a contiguous subsequence of frames extracted from a given spatio-temporal sequence for the purpose of efficient temporal modeling. Formally, let $U_j = \{\boldsymbol{x}^i\}_{t_j}^{t_j+\Delta t-1}$, where $t_j$ is the starting time of the segment and $\Delta t$ is the length of the segment measured in time units or number of frames.*

By focusing on temporal segments, we aim to capture essential temporal features while mitigating spatio-temporal redundancies that often occur in short-term sequences.

**Definition 3.2** (micro-temporal scale). *The micro-temporal scale refers to the granularity at which a spatio-temporal sequence is partitioned into non-overlapping, contiguous temporal segments for the purpose of efficient and localized temporal modeling. Formally, a sequence $\{\boldsymbol{x}^i\}_{t-T+1}^{t+T'}$ is divided into $N$ micro-temporal segments $\mathcal{U} = \{U_1, U_2, ..., U_N\}$.*

By operating at the micro-temporal scale, each temporal segment is considered an independent unit for localized temporal analysis, capturing fine-grained temporal features while avoiding overlap between different temporal segments.

**Definition 3.3** (macro-temporal scale). *The macro-temporal scale refers to the granularity at which a spatio-temporal sequence is divided into larger contiguous segments, each encompassing multiple non-overlapping micro-temporal segments. Formally, a sequence $\{\boldsymbol{x}^i\}_{t-T+1}^{t+T'}$ is divided into $M$ macro-temporal segments $\mathcal{V} = \{V_1, V_2, ..., V_M\}$, where each segment $V_j = \{U_{j1}, U_{j2}, ..., U_{jk}\}$ consists of $k$ micro-temporal segments, and $k\Delta t = \Delta T$ with $\Delta T = T \gg \Delta t$.*

For the purpose of modeling, each macro-temporal segment aims to encompass a comprehensive view of historical frames to capture the global patterns of the spatio-temporal sequence.

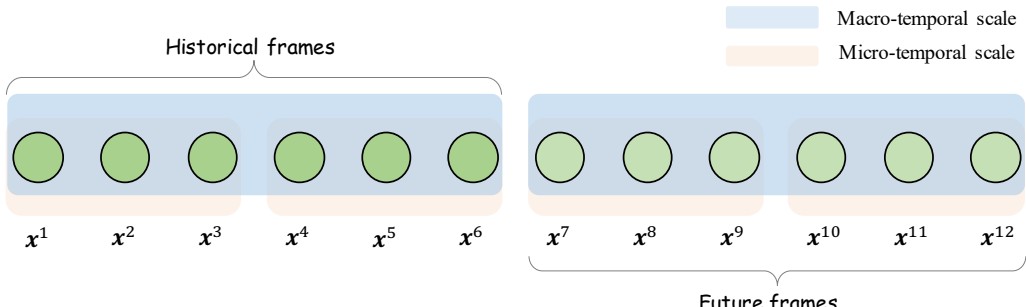

Figure 1: The illustration of micro- and macro-temporal scales. Here we take a $6 \rightarrow 6$ frames prediction as an example. Each green circle represents an individual frame. Micro-temporal scales (*in red*) divide the sequence into non-overlapping temporal segments, containing a few consecutive frames. Macro-temporal scale (*in blue*) further divides the sequence into larger temporal segments.

**Recurrent-based Temporal Modeling**   In recurrent-based temporal modeling Shi et al. (2015); Wang et al. (2017); Guen & Thome (2020), the focus is solely on micro-temporal scales, neglecting the macro-temporal scale. Formally, each micro-temporal segment $U_i$ consists solely of a single frame $\{x^i\}$ with $\Delta t = 1$. The modeling approach can be mathematically expressed as follows:

$$\widehat{U}_{i+1} = \begin{cases} \mathcal{F}_\Theta(U_i, H_{i-1}), & \text{if } t - T + 1 \leq i \leq t, \\ \mathcal{F}_\Theta(\widehat{U}_i, H_{i-1}), & \text{otherwise,} \end{cases} \quad (2)$$

where $H_{i-1}$ is the hidden state from the preceding frame. The model operates in two distinct phases:

- *Reconstruction Phase:* For historical frames $(t - T + 1 \leq i \leq t)$, the model learns to reconstruct the next frame $\widehat{U}_{i+1}$ based on the current ground-truth frame $U_i$ and the hidden state $H_{i-1}$ from the preceding frame.
- *Prediction Phase:* For future frames $i > t$, the model uses the hidden state $H_{i-1}$ and the last predicted frame $\widehat{U}_i$ to predict the next frame $\widehat{U}_{i+1}$.

In both phases, the efficacy of the model for either reconstructing or predicting frames is contingent upon the effective learning of the hidden state $H_{i-1}$ from the preceding frame.

**Recurrent-free Temporal Modeling**   In recurrent-free temporal modeling Gao et al. (2022a); Tan et al. (2023a), the focus shifts entirely to the macro-temporal scale, bypassing any micro-temporal segments. Specifically, each macro-temporal segment $V$ is defined as a sequence of $T$ consecutive frames, with $\Delta T = T$. The modeling approach can be mathematically expressed as follows:

$$\widehat{V}_2 = \mathcal{F}_\Theta(V_1), \quad (3)$$

where $V_1 = \{x^i\}_{t-T+1}^t$ is the historical frames, and $V_2 = \{x^i\}_{t+1}^{t+T'}$ is the ground-truth future frames, $\widehat{V}_2$ is the predicted future frames by the model $\mathcal{F}_\Theta$. The model operates in a single phase, where the model learns to predict the future frames $\widehat{V}_2$ based on the historical frames $V_1$. It is worth noting that here the output frames have the same length as the input frames.

By working with macro-temporal segments, recurrent-free temporal modeling exhibits computational advantages, as it can process multiple frames in parallel. It excels in capturing global patterns over the entire temporal window by taking the macro-temporal segment into account. However, it falters in handling intricate temporal dependencies, primarily because it lacks micro-temporal granularity. The fixed size of macro-temporal segments is inflexible and limits the practical applicability.

**Summary**   In summary, we have dissected the core principles of temporal modeling in both recurrent-based and recurrent-free methods. They are primarily differentiated by their focus on micro-temporal and macro-temporal scales, respectively. Figure 2 provides a comparative illustration that encapsulates the essential differences and unique characteristics of these two approaches.

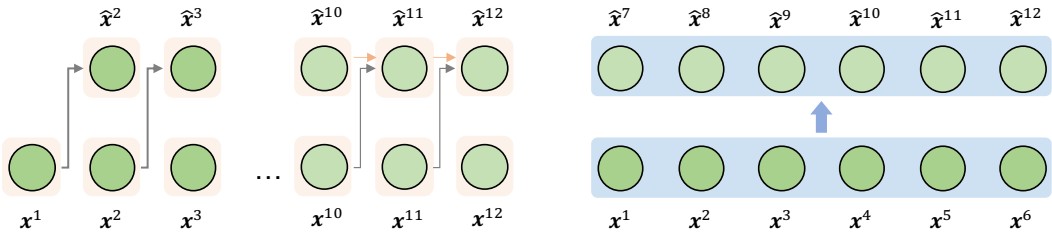

(a) Recurrent-based temporal modeling      (b) Recurrent-free temporal modeling

Figure 2: Temporal modeling comparison between recurrent-based and recurrent-free methods, illustrated using a $6 \rightarrow 6$ frames prediction example. For convenience, in (a) we use thick grey arrows to represent $\widehat{U}_i$ and $H_{i-1}$ for historical frames, while in future frames, we use thin gray arrows to indicate $H_{i-1}$ and light arrows to denote $\widehat{U}_i$.

# 4 USTEP: UNIFIED SPATIO-TEMPORAL PREDICTIVE LEARNING

To overcome the limitations inherent in both recurrent-based and recurrent-free temporal modeling methods, we propose USTEP, a unified framework designed to harmoniously integrate micro- and macro-temporal scales. It aims to maximize the effectiveness of spatio-temporal predictive learning.

## 4.1 TEMPORAL SCALE SETS

The initial stage in USTEP involves dividing the input frame sequence into two separate sets, corresponding to the micro-temporal and macro-temporal scales, which are denoted as $\mathcal{U}$ and $\mathcal{V}$, respectively. In the micro-temporal scale set $\mathcal{U}$, each temporal segment $U_i$ is constructed to contain a few consecutive frames, facilitating the capture of fine-grained spatio-temporal information. The length of each $U_i$ is determined by $\Delta t$, which is chosen to balance the trade-off between temporal granularity and computational efficiency. For the macro-temporal scale set $\mathcal{V}$, we employ a sliding window approach to construct larger temporal segments. Each macro-temporal segment $V_i$ contains multiple non-overlapping segments $U_i$. The sliding window moves in steps of size $\Delta t$, ensuring that the macro-temporal segments are constructed in a manner consistent with the micro-temporal scale. We show the illustration of micro- and macro-temporal scale sets of USTEP in Figure 3.

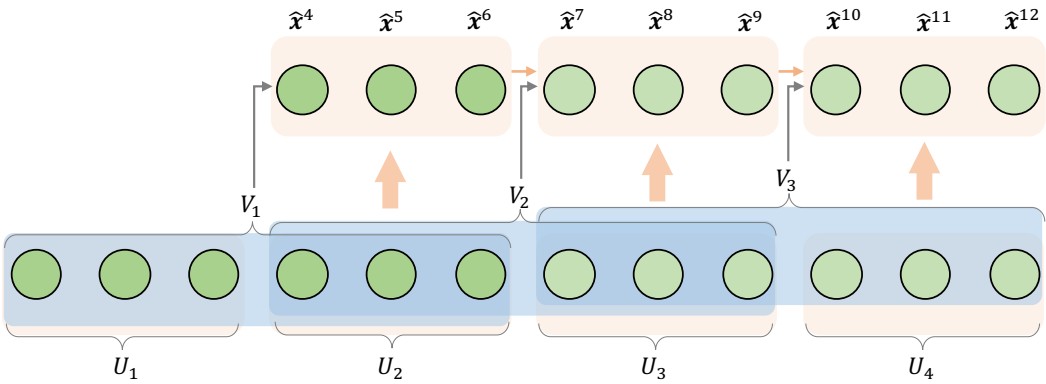

Figure 3: Schematic representation of the micro- and macro-temporal scale sets in USTEP. In this example, we use $\Delta t = 3$ to construct the micro-temporal segments $U_i$, and $\Delta T = 6$ to construct the macro-temporal segments $V_i$. Analogous to recurrent-based approaches, the first micro-temporal segment is not utilized for predictions while learning temporal dependencies in subsequent frames.

## 4.2 SINGLE SEGMENT-LEVEL TEMPORAL MODELING

Upon segmenting the input frame sequence into the micro- and macro-temporal sets, $\mathcal{U}$ and $\mathcal{V}$, the next step is to perform temporal modeling at the single-segment level. The primary goal of this stage is to obtain compatible hidden states for both temporal scales while ensuring they are mapped to a uniform feature space with the same dimension in a recurrent-free manner.

To accomplish this, we map the temporal segments $U_i$ and $V_i$ from both micro- and macro-scales to a unified feature space with dimensions $C' \times H \times W$. Specifically, we utilize two distinct recurrent-free modules, $F_{\theta_1}^U$ and $F_{\theta_2}^V$, each parameterized by learnable parameters $\theta_1, \theta_2$. We implement them using TAU Tan et al. (2023a) modules in practice. These modules transform the original segments to corresponding hidden states, denoted as $\overline{U}_i, \overline{V}_i \in \mathbb{R}^{C' \times H \times W}$, according to the following equations:

$$\overline{U}_i = F_\theta^U(U_i), \quad \overline{V}_i = F_\theta^V(V_i), \tag{4}$$

By mapping both micro- and macro-temporal segments to the same feature dimension $C'$, we establish a unified representation space where the hidden states from both temporal scales can be directly integrated. Such a unified feature space enables us to leverage the complementary strengths of micro- and macro-temporal modeling, thus enhancing the overall performance.

### 4.3 CROSS SEGMENT-LEVEL TEMPORAL MODELING

Once the unified hidden states are obtained from both micro- and macro-temporal segments, the next challenge is to harmoniously integrate these at the cross-segment level. Our approach is to leverage the advantages of both micro- and macro-temporal scales, capturing fine-grained detail while maintaining a global perspective, to enhance the predictive capability of our model, USTEP.

The macro-temporal scale hidden states, $\overline{V}_i$, are processed using a gating mechanism as follows:

$$
\begin{aligned}
\boldsymbol{g}_i &= \sigma(W_v * \overline{V}_i + b_v), \\
\boldsymbol{h}_i^V &= \overline{V}_i + \boldsymbol{g}_i \odot \boldsymbol{h}_{i-1}^V,
\end{aligned}
\tag{5}
$$

where $\sigma(\cdot)$ denotes the Sigmoid activation function, $*$ and $\odot$ represent the convolution operator and the Hadamard product, respectively. The parameters $W_v$ and $b_v$ are the convolution weights and bias, and $\boldsymbol{g}_i$ is the gate controlling the flow of historical macro-temporal scale information.

Subsequently, the micro-temporal scale hidden state and the processed macro-temporal hidden state are integrated as follows:

$$
\begin{aligned}
\boldsymbol{m}_i &= \sigma(W_u * \overline{U}_i + b_u), \\
\boldsymbol{c}_i &= \sigma(W_c * \overline{U}_i + b_c), \\
\boldsymbol{h}_i^U &= \overline{U}_i + \boldsymbol{m}_i \odot \boldsymbol{h}_{i-1}^U + \boldsymbol{c}_i \odot \boldsymbol{h}_{i-1}^V,
\end{aligned}
\tag{6}
$$

where $W_u, W_c, b_u$, and $b_c$ are the convolution weights and biases. The historical gate $\boldsymbol{m}_i$ and the cross-segment gate $\boldsymbol{c}_i$ modulate the integration of historical micro-temporal scale information $\boldsymbol{h}_{i-1}^U$ and the preceding macro-temporal segment hidden state $\boldsymbol{h}_{i-1}^V$.

The overall schematic diagram of temporal modeling learning in USTEP is shown in Figure 3. For every single segment, regardless of whether it belongs to the micro-temporal or macro-temporal scale, a recurrent-free approach is employed to swiftly capture coarse temporal dependencies. In contrast, when considering the relationships across segments, a recurrent-based approach is sequentially applied to discern finer temporal dependencies. Notably, during this stage, temporal dependencies from different scales are harmoniously fused, ensuring a comprehensive temporal understanding. This hierarchical and integrative approach allows USTEP to achieve a delicate balance between capturing immediate temporal nuances and understanding broader temporal patterns.

---

**Algorithm 1** Pseudocode of train

```
def train(data, delta_t, delta_T):
    x = data[:, :-delta_t]
    y = data[:, delta_t:]
    # partition: micro and macro sets
    u, v = partition(x, delta_t, delta_T)

    # single segment-level: recurrent-free
    u, v = f_u(u), f_v(v)
    # cross segment-level: recurrent-based
    pred = []
    h_v, h_u = 0, 0
    for i in range(len(u)):
        h_v = macro_func(v[i], h_v)
        h_u = micro_func(u[i], h_u, h_v)
        pred.append(h_u)

    loss = loss_func(pred, y)
    return loss
```

**Algorithm 2** Pseudocode of inference

```
def inference(x, delta_t, delta_T, n_step):
    # x: B x T x C x H x W
    # partition: micro and macro sets
    u, v = partition(x, delta_t, delta_T)

    # single segment-level: recurrent-free
    u, v = f_u(u), f_v(v)
    # cross segment-level: recurrent-based
    pred = []
    h_v, h_u = 0, 0
    for i in range(n_step):
        h_v = macro_func(v[i], h_v)
        h_u = micro_func(u[i], h_u, h_v)
        # practical prediction
        if i >= len(u):
            pred.append(h_u)

    return pred
```

---

The pseudocode for USTEP is delineated in Algorithm 1 for the training phase and Algorithm 2 for the inference phase. In the training phase, our method adopts a partitioning approach to divide the input sequences into $x$ and $y$, which aligns with the training strategy typically employed by recurrent-based methods. However, while traditional methods are rigid, employing a $\Delta t = 1$, our approach offers flexibility. During the inference phase, the model engages in iterative operations for a predetermined number of steps. These steps encompass both those required for observing sequences $x$ and the actual steps required for generating the predictions.

# 5 EXPERIMENTS

We evaluate the efficacy of USTEP across three prevalent types of spatiotemporal prediction tasks:

- *Equal Frame Task:* The number of output frames matches that of the input frames. This task type inherently favors recurrent-free temporal modeling approaches due to its structured nature.

- *Extended Frame Task:* The count of output frames substantially surpasses that of the input frames. This type of task is generally more compatible with recurrent-based temporal modeling approaches, allowing for more flexible, frame-by-frame predictions.

- *Reduced Frame Task:* Diverging from the former, this task necessitates fewer output frames than the input frames. By mitigating the impact of cumulative errors, this task directly evaluates the model's capability in learning historical frames.

The details about experimental settings and visualizations are shown in Appendix A and C. The choices of $\Delta t$ and $\Delta T$ are discussed in Appendix B.

## 5.1 EQUAL FRAME TASK

Under the experimental setup of the Equal frame task, we evaluated the performance of the model on three datasets: Moving MNIST, Human3.6M, and WeatherBench. For these datasets, the tasks are to predict 10 frames from 10 frames, 4 frames from 4 frames, and 12 frames from 12 frames.

Table 1: The results on the equal frame task. The units for Params and FLOPs are M and G.

| Dataset | Metric | Method | | | | | |
|---|---|---|---|---|---|---|---|
| **Moving MNIST** | | ConvLSTM | PredRNN | PredRNN++ | MIM | E3DLSTM | PredRNNv2 |
| | MSE | 29.80 | 23.97 | 22.06 | 22.55 | 35.97 | 24.13 |
| | MAE | 90.64 | 72.82 | 69.58 | 69.97 | 78.28 | 73.73 |
| | SSIM ($\times 10^{-2}$) | 92.88 | 94.62 | 95.09 | 94.98 | 93.20 | 94.53 |
| | PSNR | 22.10 | 23.28 | 23.65 | 23.56 | 21.11 | 23.21 |
| | Params | **15.0** | 23.8 | 38.6 | 38.0 | 51.0 | 23.9 |
| | FLOPs | 56.8 | 116.0 | 171.7 | 179.2 | 298.9 | 116.6 |
| | | SimVP | TAU | Uniformer | MLP-Mixer | ConvNext | USTEP |
| | MSE | 32.15 | 24.60 | 30.38 | 29.52 | 26.94 | **21.84** |
| | MAE | 89.05 | 71.93 | 85.87 | 83.36 | 77.23 | **63.21** |
| | SSIM ($\times 10^{-2}$) | 92.68 | 94.54 | 93.08 | 93.38 | 93.97 | **95.38** |
| | PSNR | 21.84 | 23.19 | 22.78 | 22.13 | 22.22 | **24.06** |
| | Params | 58.0 | 44.7 | 46.8 | 44.8 | 37.3 | 18.9 |
| | FLOPs | 19.4 | 16.0 | 16.5 | 16.5 | **14.1** | 17.7 |
| **Human3.6M** | | ConvLSTM | PredRNN | PredRNN++ | MIM | E3DLSTM | PredRNNv2 |
| | MSE | 125.5 | 113.2 | 110.0 | 112.1 | 143.3 | 114.9 |
| | MAE | 1566.7 | 1458.3 | 1452.2 | 1467.1 | 1442.5 | 1484.7 |
| | SSIM ($\times 10^{-2}$) | 98.13 | 98.31 | 98.32 | 98.29 | 98.03 | 98.27 |
| | PSNR | 33.40 | 33.94 | 34.02 | 33.97 | 32.52 | 33.84 |
| | Params | 15.5 | 24.6 | 39.3 | 47.6 | 60.9 | 24.6 |
| | FLOPs | 347.0 | 704.0 | 1033.0 | 1051.0 | 542.0 | 708.0 |
| | | SimVP | TAU | Uniformer | MLP-Mixer | ConvNext | USTEP |
| | MSE | 115.8 | 113.3 | **108.4** | 116.3 | 113.4 | 109.5 |
| | MAE | 1511.5 | 1390.7 | 1441.0 | 1497.7 | 1469.7 | **1380.5** |
| | SSIM ($\times 10^{-2}$) | 98.22 | 98.39 | 98.34 | 98.24 | 98.28 | **98.45** |
| | PSNR | 33.73 | 34.03 | 34.08 | 33.76 | 33.86 | **34.35** |
| | Params | 41.2 | 37.6 | 11.3 | 27.7 | 31.4 | **3.7** |
| | FLOPs | 197.0 | 182.0 | 74.6 | 211.0 | 157.0 | **66.2** |
| **WeatherBench** | | ConvLSTM | PredRNN | PredRNN++ | MIM | E3DLSTM | PredRNNv2 |
| | MSE | 1.521 | 1.331 | 1.634 | 1.784 | 1.592 | 1.545 |
| | MAE ($\times 10^{-2}$) | 79.49 | 72.46 | 78.83 | 87.16 | 80.59 | 79.86 |
| | RMSE | 1.233 | 1.154 | 1.278 | 1.336 | 1.262 | 1.243 |
| | Params | 14.98 | 23.57 | 38.31 | 37.75 | 51.09 | 23.59 |
| | FLOPs | 136 | 278 | 413 | 109 | 169 | 279 |
| | | SimVP | TAU | Uniformer | MLP-Mixer | ConvNext | USTEP |
| | MSE | 1.238 | 1.162 | 1.204 | 1.255 | 1.277 | **1.150** |
| | MAE ($\times 10^{-2}$) | 70.37 | 67.07 | 68.85 | 70.11 | 72.20 | **65.83** |
| | RMSE | 1.113 | 1.078 | 1.097 | 1.119 | 1.130 | **1.072** |
| | Params | 14.67 | 12.22 | 12.02 | 11.10 | 10.09 | **3.59** |
| | FLOPs | 8.03 | **6.70** | 7.45 | 5.92 | 5.66 | 8.2 |

The experimental results are summarized in the Table 1. In the Moving MNIST dataset, USTEP outshines the top-performing recurrent-based model, PredRNN++, achieving superior results while utilizing only half the Params and a tenth of the FLOPs. Concurrently, compared to the leading recurrent-free model, TAU, USTEP operates with comparable FLOPs, approximately 42% of its parameters, and demonstrates substantial enhancements in performance metrics. On the Human3.6M dataset, USTEP demonstrates the best performance in all metrics except for MSE. Notably, it employs one-third of the parameters of the second-best model, Uniformer, and executes approximately 12% fewer FLOPs. On WeatherBench, USTEP yields impressive results, securing considerable advancements across various metrics. It maintains FLOPs close to the premier recurrent-free model, TAU, while its parameter count is merely 29% of TAU's. Moreover, compared to the leading recurrent-based model, PredRNN, the parameters of USTEP are just 15% of it, underscoring USTEP's efficiency and effectiveness in spatiotemporal predictive learning tasks.

## 5.2 EXTENDED FRAME TASK

For this task, we conduct evaluations using the KTH dataset, with the results detailed in Table 2. This task requires the model to adeptly predict a greater number of frames than the number of observed frames. A cursory observation reveals that, while recurrent-free models exhibit efficiency in both Params and FLOPs, they lag considerably in performance when juxtaposed with the top recurrent-based model, PredRNN++. USTEP stands out by eclipsing the performance of PredRNN++ and concurrently preserving an efficiency level in line with that of the recurrent-free models.

Table 2: Quantitative results of different methods on the KTH dataset ($10 \rightarrow 20$ frames).

| Method | Params (M) | FLOPs (G) | MSE ↓ | MAE ↓ | SSIM ↑ | PSNR ↑ | LPIPS ↓ |
|---|---|---|---|---|---|---|---|
| ConvLSTM | 14.9 | 1368.0 | 47.65 | 445.5 | 0.8977 | 26.99 | 0.26686 |
| PredRNN | 23.6 | 2800.0 | 41.07 | 380.6 | 0.9097 | 27.95 | 0.21892 |
| PredRNN++ | 38.3 | 4162.0 | _39.84_ | 370.4 | _0.9124_ | _28.13_ | _0.19871_ |
| MIM | 39.8 | 1099.0 | 40.73 | 380.8 | 0.9025 | 27.78 | **0.18808** |
| E3DLSTM | 53.5 | 217.0 | 136.40 | 892.7 | 0.8153 | 21.78 | 0.48358 |
| PredRNNv2 | 23.6 | 2815.0 | 39.57 | _368.8_ | 0.9099 | 28.01 | 0.21478 |
| SimVP | _12.2_ | **62.8** | 41.11 | 397.1 | 0.9065 | 27.46 | 0.26496 |
| TAU | 15.0 | 73.8 | 45.32 | 421.7 | 0.9086 | 27.10 | 0.22856 |
| Uniformer | **11.8** | 78.3 | 44.71 | 404.6 | 0.9058 | 27.16 | 0.24174 |
| MLP-Mixer | 20.3 | 66.6 | 57.74 | 517.4 | 0.8886 | 25.72 | 0.28799 |
| ConvNext | 12.5 | _63.9_ | 45.48 | 428.3 | 0.9037 | 26.96 | 0.26253 |
| USTEP | 12.8 | 107.0 | **39.55** | **364.9** | **0.9165** | **28.98** | 0.19956 |

## 5.3 REDUCED FRAME TASK

We aim to assess the models' proficiency in learning from observed frames while minimizing accumulated errors. The evaluation is performed using the Caltech Pedestrian dataset, and the derived results are encapsulated in Table 3. USTEP notably outperforms the recurrent-based models, achieving superior results with a minimal parameter footprint and lower computational overhead.

Table 3: Quantitative results of different methods on the Caltech Pedestrian dataset ($10 \rightarrow 1$ frame).

| Method | Params (M) | FLOPs (G) | MSE ↓ | MAE ↓ | SSIM ↑ | PSNR ↑ | LPIPS ↓ |
|---|---|---|---|---|---|---|---|
| ConvLSTM | 15.0 | 595.0 | 139.6 | 1583.3 | 0.9345 | 27.46 | 8.58 |
| PredRNN | 23.7 | 1216.0 | 130.4 | 1525.5 | 0.9374 | 27.81 | 7.40 |
| PredRNN++ | 38.5 | 1803.0 | 125.5 | _1453.2_ | 0.9433 | 28.02 | 13.21 |
| MIM | 49.2 | 1858.0 | _125.1_ | 1464.0 | 0.9409 | _28.10_ | 6.35 |
| E3DLSTM | 54.9 | 1004.0 | 200.6 | 1946.2 | 0.9047 | 25.45 | 12.60 |
| PredRNNv2 | 23.8 | 1223.0 | 147.8 | 1610.5 | 0.9330 | 27.12 | 8.92 |
| SimVP | _8.6_ | **60.6** | 160.2 | 1690.8 | 0.9338 | 26.81 | 6.76 |
| TAU | 15.0 | 92.5 | 131.1 | 1507.8 | _0.9456_ | 27.83 | _5.49_ |
| Uniformer | 11.8 | 104.0 | 135.9 | 1534.2 | 0.9393 | 27.66 | 6.87 |
| MLP-Mixer | 22.2 | 83.5 | 207.9 | 1835.9 | 0.9133 | 26.29 | 7.75 |
| ConvNext | 12.5 | _80.2_ | 146.8 | 1630.0 | 0.9336 | 27.19 | 6.99 |
| USTEP | **5.1** | 85.3 | **123.6** | **1407.9** | **0.9477** | **28.37** | **4.94** |

## 5.4 Ablation Study

To further comprehend the impact of different design choices on the performance of USTEP, we conducted an extensive ablation study. This study emphasizes the influence of varying the temporal stride $\Delta t$ and the role of the cross-segment mechanism within the model. From the Table 4, it is observed that USTEP, with $\Delta t = 5$, achieves the lowest MSE of 21.84 and the highest SSIM of 0.9538. The results also highlight that reducing $\Delta t$ leads to an increase in FLOPs due to the more frequent computations required, impacting the model's efficiency. Moreover, the cross-segment mechanism is shown to be crucial, as its removal leads to a significant drop in performance.

Table 4: Ablation study on the influence of different design choices on the Moving MNIST dataset.

| Method | Params (M) | FLOPS (G) | MSE↓ | MAE↓ | SSIM↑ | PSNR↑ |
|---|---|---|---|---|---|---|
| | | | Moving MNIST | | | |
| USTEP ($\Delta t = 5$) | 18.9 | 17.7 | **21.84** | 63.21 | **0.9538** | 24.06 |
| USTEP ($\Delta t = 1$) | 18.8 | 52.5 | 31.94 | 71.72 | 0.9416 | 23.36 |
| USTEP ($\Delta t = 2$) | 18.3 | 30.8 | 24.62 | **62.30** | 0.9525 | **24.46** |
| USTEP ($\Delta t = 10$) | 19.1 | 13.4 | 25.13 | 74.31 | 0.9440 | 23.02 |
| USTEP w/o cross segment | 17.4 | 13.1 | 24.01 | 67.65 | 0.9489 | 23.57 |

Figure 4 delineates that $\Delta t = 1$ tends to overemphasize local information, potentially leading to a lack of holistic understanding. In contrast, $\Delta t = 10$ appears to overly prioritize global information, possibly at the expense of missing finer, localized details. These insights underline the critical importance of choosing an appropriate $\Delta t$ in USTEP to balance local and global spatiotemporal considerations, ensuring the holistic integrity of the learned features and predictions.

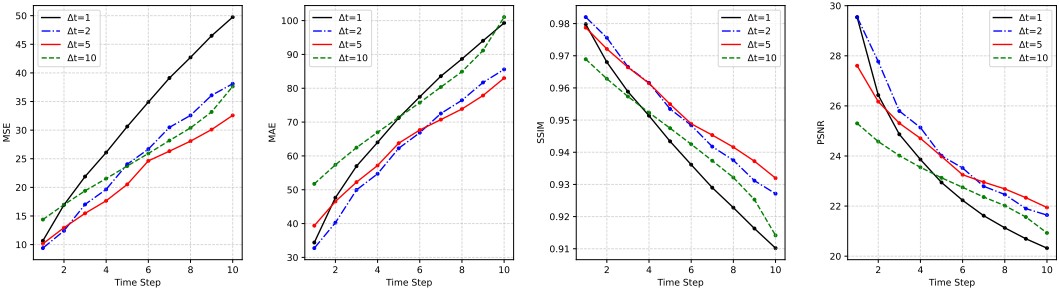

Figure 4: Frame-wise comparison in MSE, MAE, SSIM and PSNR metrics. For MSE and MAE, lower values are preferable. For SSIM and PSNR, higher values are more desirable.

## 6 Conclusion and Limitation

This paper introduced USTEP, a novel paradigm for spatiotemporal prediction tasks, thoughtfully architected to unify the strengths of both recurrent-based and recurrent-free models. USTEP operates under a novel paradigm that offers a comprehensive view of spatiotemporal dynamics, facilitating a nuanced understanding and representation of intricate temporal patterns and dependencies. USTEP has proven its mettle across a variety of spatiotemporal tasks, demonstrating exceptional adaptability and superior performance in diverse contexts. It meticulously integrates local and global spatiotemporal information, providing a unified perspective that enhances its performance.

While USTEP has showcased promising results across various spatiotemporal tasks, its efficacy is inherently contingent on the selection of the temporal stride $\Delta t$. Adaptively adjusting the temporal stride based on the characteristics of the input sequence could be valuable. Further, the performance might be constrained in scenarios with highly irregular and unpredictable temporal patterns.

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

## A  EXPERIMENTAL SETTINGS AND DATASETS

**Datasets**  We quantitatively evaluate our model on the following datasets for both synthetic and real-world scenarios:

- **Moving MNIST** Srivastava et al. (2015) is a synthetic dataset consisting of two digits independently moving within the $64 \times 64$ grid and bouncing off the boundary. It is a standard benchmark in spatiotemporal predictive learning.

- **Human 3.6M** Ionescu et al. (2013) is a large-scale 3D human motion capture dataset for fitness, close human interactions, and self-contact. This dataset contains 3.6 million 3D human poses and corresponding images, 11 professional actors (6 male, five female), and 17 scenarios (discussion, smoking, taking photos, talking on the phone, etc.).

- **Weather Benchmark** Rasp et al. (2020) This dataset contains various types of climatic data from 1979 to 2018. The raw data is regrind to low resolutions, we here choose $5.625°$ ($32 \times 64$ grid points) resolution for our data. Since the complete data is very large and includes massive climatic attributes like geopotential, temperature, and other variables, we specifically chose the global temperature prediction task to evaluate our model.

- **Caltech Pedestrian** is a driving dataset focusing on detecting pedestrians. It consists of approximately 10 hours of $640 \times 480$ videos taken from vehicles driving through regular traffic in an urban environment. We follow the same protocol of PredNet Lotter et al. (2017) and CrevNet Yu et al. (2019) for pre-processing, training, and evaluation.

- **KTH** Schuldt et al. (2004) contains 25 individuals performing six types of actions. Following Villegas et al. (2017); Wang et al. (2018b), we use persons 1-16 for training and 17-25 for testing. Models are trained to predict the next 20 frames from the previous 10 observations.

We summarize the statistics of the above datasets in Table 5, including the number of training samples $N_{train}$ and the number of testing samples $N_{test}$.

Table 5: The statistics of datasets. The training or testing set has $N_{train}$ or $N_{test}$ samples, composed by $T$ or $T'$ images with the shape $(C, H, W)$.

|            | $N_{train}$ | $N_{test}$ | $(C, H, W)$    | $T$ | $T'$ |
|------------|-------------|------------|----------------|-----|------|
| MMNIST     | 10,000      | 10,000     | (1, 64, 64)    | 10  | 10   |
| Human 3.6M | 73,404      | 8,582      | (3, 256, 256)  | 4   | 4    |
| WeatherBench | 2,167     | 706        | (1, 32, 64)    | 12  | 12   |
| Kitti&Caltech | 3,160    | 3,095      | (3, 128, 160)  | 10  | 1    |
| KTH        | 4,940       | 3,030      | (1, 128, 128)  | 10  | 20   |

**Baselines**  We choose the following baselines for comparison: (i) Recurrent-based methods including ConvLSTM Shi et al. (2015), PredRNN Wang et al. (2017), PredRNN++ Wang et al. (2018a), MIM Wang et al. (2019a), E3D-LSTM Wang et al. (2018b), and PredRNNv2 Wang et al. (2021); (ii) Recurrent-free methods including SimVP Gao et al. (2022a), TAU Tan et al. (2023a), Uniformer Li et al. (2022), MLP-Mixer Tolstikhin et al. (2021), and ConvNeXt Liu et al. (2022).

**Measurement**  We employ Mean Squared Error (MSE), Mean Absolute Error (MAE), Structure Similarity Index Measure (SSIM), and Peak Signal to Noise Ratio (PSNR) to evaluate the quality of predictions. MSE and MAE estimate the absolute pixel-wise errors, SSIM measures the similarity of structural information within the spatial neighborhoods, and PSNR is an expression for the ratio between the maximum possible power of a signal and the power of distorted noise. LPIPS Zhang et al. (2018) is a perceptual similarity metric that computes the distance between two images' feature representations in a pre-trained deep network.

**Implementation details**  We implement the proposed method with the Pytorch framework and conduct experiments on a single NVIDIA-V100 GPU. The AdamW optimizer is utilized with a learning rate of 0.01 and a weight decay of 0.05.

# B DISCUSSION ABOUT THE TEMPORAL STRIDE

The choice of the temporal stride $\Delta t$ plays a crucial role in navigating the trade-off between performance and efficiency, thereby impacting the performance of USTEP in various spatiotemporal prediction tasks. In contrast, $\Delta T$ is set to be the same as $T$ for all datasets to capture the macro-temporal scale dependencies. Table 6 outlines the selected values for $\Delta t$ across different datasets, providing insights into the alignment of the model's focus with the inherent characteristics.

Table 6: The choices of $\Delta t$ and $\Delta T$ for different datasets.

|  | MMNIST | Human 3.6M | WeatherBench | Caltech | KTH |
|---|---|---|---|---|---|
| $T$ | 10 | 4 | 12 | 10 | 10 |
| $T'$ | 10 | 4 | 12 | 1 | 20 |
| $\Delta t$ | 5 | 2 | 4 | 5 | 10 |
| $\Delta T$ | 10 | 4 | 12 | 10 | 10 |

The value of $\Delta t$ is chosen to be half of $T$ for MMNIST, reflecting a balanced approach to incorporating both micro- and macro-temporal scale information. This approach is mirrored in Caltech dataset as well, with $\Delta t$ being half of $T$ to ensure the synthesis of local and global perspectives. For Human 3.6M, a smaller $\Delta t$ is selected to give more weight to micro-temporal scale dependencies, given the dataset's nuanced temporal dynamics. Similarly, in WeatherBench, a $\Delta t$ of 4 is chosen to provide a balanced view of the temporal sequence, catering to the dataset's diverse temporal patterns. For the KTH dataset, $\Delta t$ equals $T$ as the dataset is relatively simple, allowing the model to harness the whole input temporal context within only one micro-temporal segment to generate coherent and plausible future frames.

## C VISUALIZATION

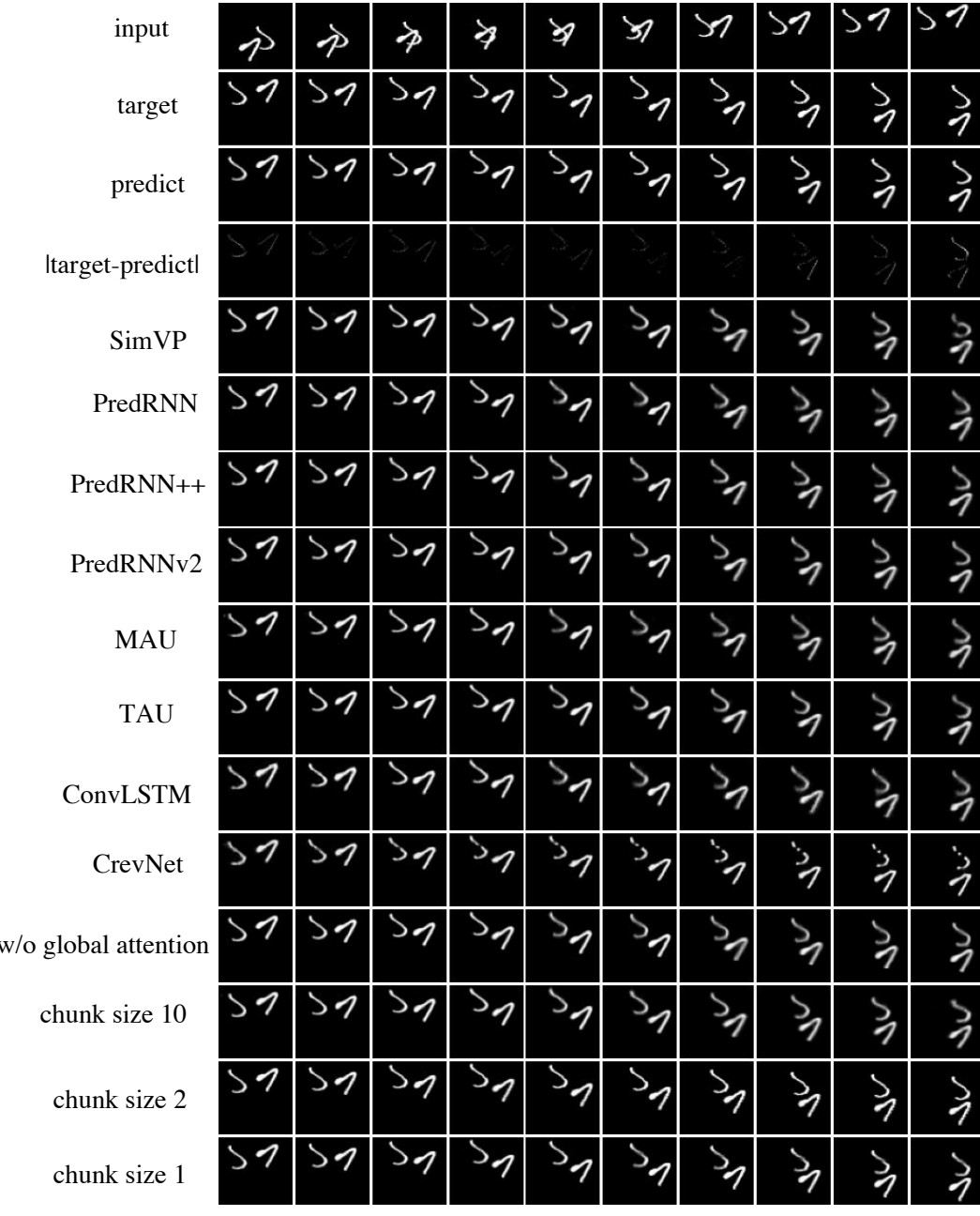

Figure 5: The qualitative visualization on Moving MNIST.

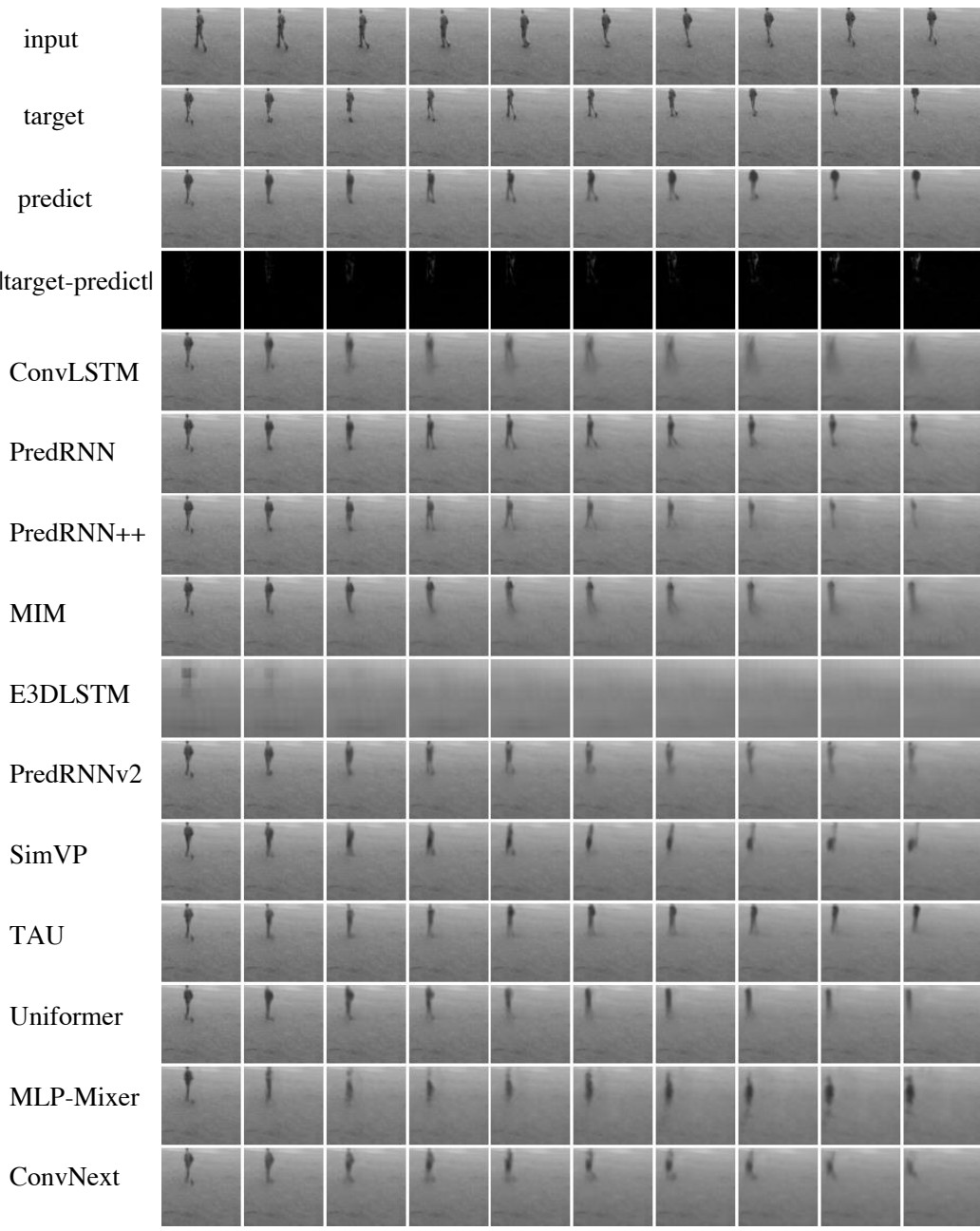

Figure 6: The qualitative visualization on KTH dataset. The target and predicted sequence is the range of {2,4,...,20}

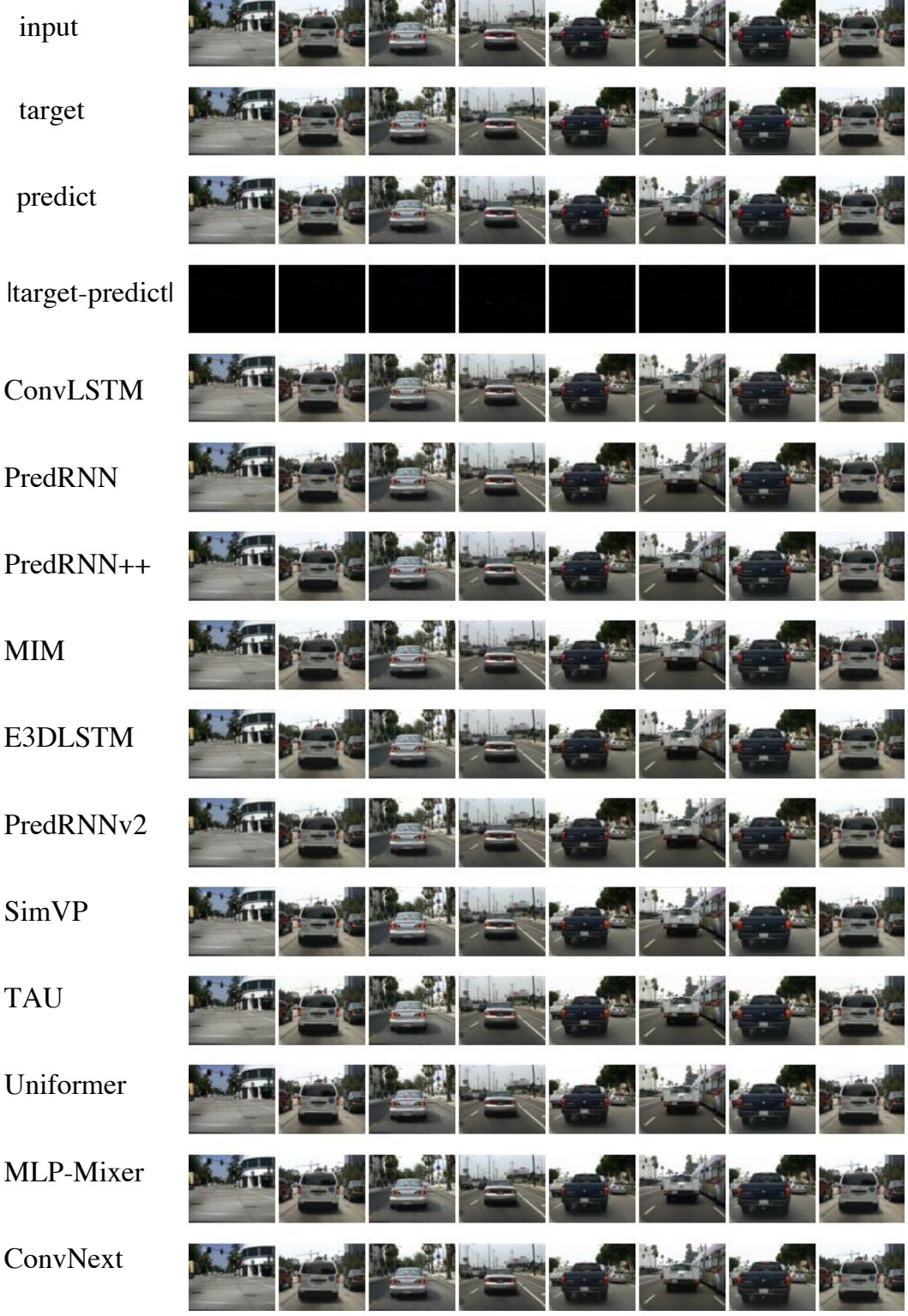

Figure 7: The qualitative visualization on Caltech Pedestrian dataset.

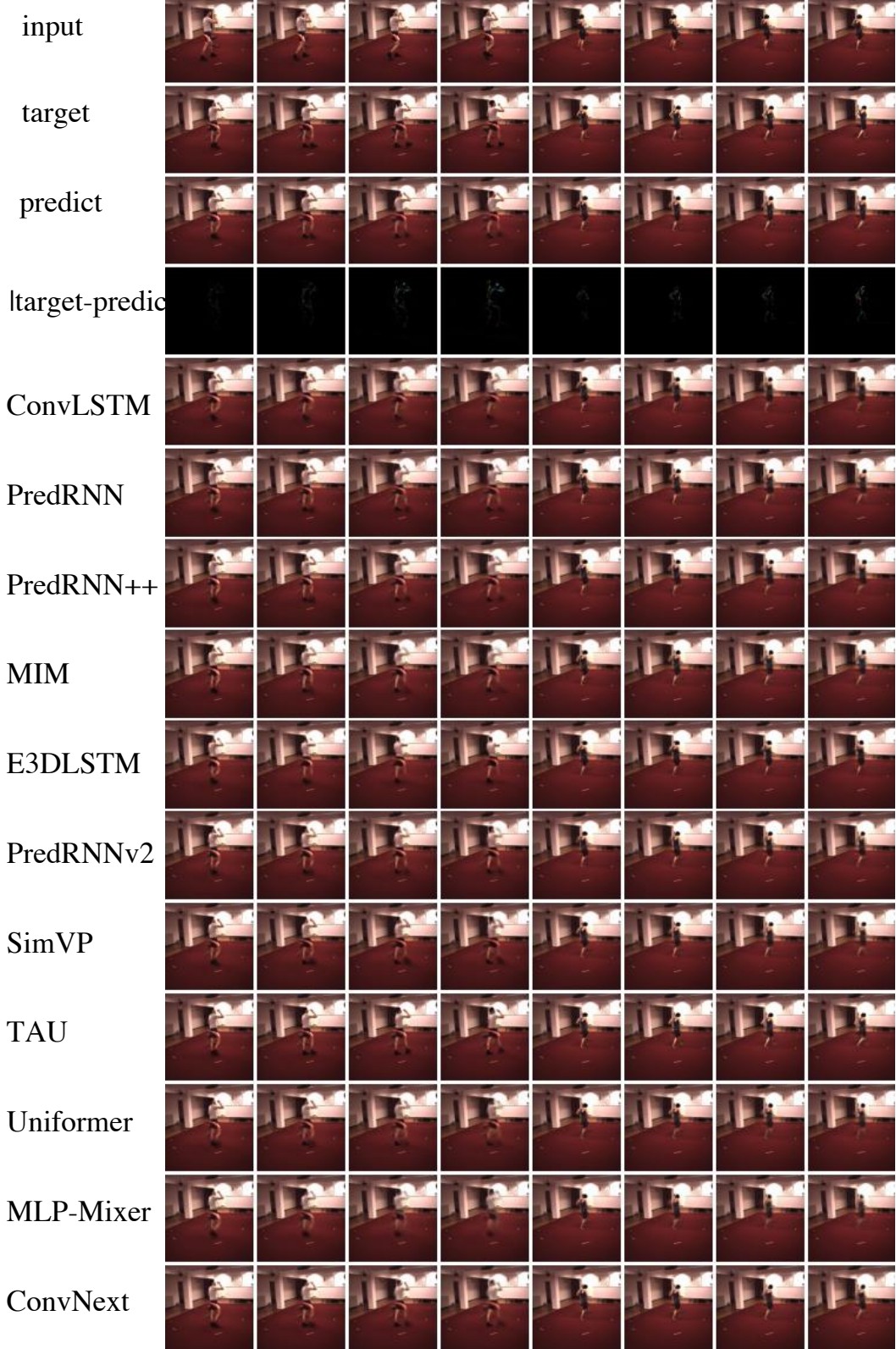

Figure 8: The qualitative visualization on Human 3.6M.

