# OpenReview forum: "Revisiting the Temporal Modeling in Spatio-Temporal Predictive Learning under A Unified View"
_ICLR.cc/2024/Conference — ICLR 2024 Conference Withdrawn Submission_

### Official Review · Reviewer_sHRf · 2023-10-20

**Soundness:** 2 fair
**Presentation:** 3 good
**Contribution:** 2 fair
**Rating:** 5
**Confidence:** 3

**Summary:**

The authors review the advantages and disadvantages of two main paradigms for temporal modeling, recurrent-based and recurrent-free. They propose a hybrid approach named USTEP to jointly capture micro-scale and macro-scale dynamics by designing a network that combines the convolution and the gating mechanisms. The experimental results of three types of spatio-temporal prediction tasks show that USTEP performs the best in most cases.

**Strengths:**

1. The writing is clear and easy to follow. Proper definitions, concise formulas, and pseudo codes increase the readability.
2. The experimental design and analysis are generally comprehensive.

**Weaknesses:**

1. My main concern is that the problem studied in this manuscript can be resolved by multi-scale temporal modeling. The granularity is decided by the length of the temporal segment. Two specific scales, i.e., the micro-scale and the macro-scale are considered in this manuscript. However, there can be more scales affecting the dynamics but not covered by USTEP. Therefore, the authors are suggested to review literature on multi-scale temporal modeling, and tell the differences between these works and USTEP. Some related papers limited to my scope are listed as follows:

[1] WaveNet: A generative model for raw audio. In CoRR, 2016.

[2] Timeseries anomaly detection using temporal hierarchical one-class network. In NeurIPS, 2020.

2. The convolution operator and the gating mechanism have been widely used in existing work. Thus, this manuscript does not seem to introduce new techniques to temporal modeling.

**Questions:**

1. How to form the pair $(U_i, V_i)$? What are the sizes of $\mathcal{U}$ and $\mathcal{V}$, respectively? It seems that $|\mathcal{U}| > |\mathcal{V}|$. Thus, padding may be required to form extra macro-temporal segments.
2. The authors empirically show the computational advantages of USTEP in terms of Params and FLOPs. Can you provide the time and space complexity analysis of USTEP, recurrent-based temporal modeling and recurrent-free temporal modeling? As I expected, the complexity of USTEP is higher than its competitors since it is a hybrid method.

Minor:
1. Figure 1-3 seem to take up too much space to convey the information. On the other hand, an illustrative figure for the pipeline of USTEP is missing.
2. The upwards and downwards arrows used in Table 2 and Table 3 increase the readability of the tables. How about marking all metrics in all tables？
3. Descriptions and analysis are missing for figures in Appendix C.

---

### Official Review · Reviewer_ygZp · 2023-10-27

**Soundness:** 3 good
**Presentation:** 3 good
**Contribution:** 3 good
**Rating:** 3
**Confidence:** 4

**Summary:**

1. Spatio-temporal predictive learning is vital in self-supervised learning, and previous temporal modeling methods can be divided into recurrent-based and recurrent-free methods, each with its shortcomings.
2. This paper revisits these two predominant temporal modeling methods, providing a unified perspective and introduces USTEP (Unified Spatio-TEmporal Predictive learning), a novel framework that blends both methods and addresses their respective limitations.
3. Through comprehensive experiments across various spatio-temporal predictive learning tasks, USTEP is shown to achieve notable enhancements over traditional temporal modeling techniques, positioning it as a sturdy solution for numerous spatio-temporal applications.

**Strengths:**

1.The theme of the paper is captivating, with a perspective that's truly intriguing.

2.The paper has conducted numerous in-depth experiments.

**Weaknesses:**

1. This study delves into a captivating topic, examining two modes of prediction. However, the experimental approach presented by the author seems more like a clever technique than a genuine academic investigation. It would be beneficial if the author could furnish theoretical substantiations, perhaps drawing inspiration from tactics in the field of numerical methods, to further discuss the accumulation of errors during recursive predictions.

2. The author posits that meticulous frame-by-frame processing might overlook short-term spatiotemporal information redundancies, leading to inefficiencies. On the other hand, simply stacking frames in chronological order might miss out on deeper temporal dependencies. Yet, I couldn't seem to find experimental data in the text supporting this view. If I missed something, please enlighten me. As far as I'm aware, Simvp employs a method of stacking frames sequentially and multiplies T with C. Their performance in the temporal dimension seems commendable, which makes me question the author's conclusions.

3. The baseline provided in this study seems somewhat lacking. I would suggest the author to consider the relevant research in references [1-2] and attempt to integrate them into the discussion.

4. In the paper, the datasets chosen by the author predominantly have equal input and output lengths. To bolster the author's stance on long-term predictive capabilities, it would be advisable to choose more challenging datasets where the prediction length far exceeds the input length.

[1] Ning, Shuliang, et al. "MIMO is all you need: a strong multi-in-multi-out baseline for video prediction." In *Proceedings of the AAAI Conference on Artificial Intelligence*, vol. 37, no. 2, pp. 1975-1983. 2023.

[2] Seo, Minseok, et al. "Implicit Stacked Autoregressive Model for Video Prediction." *arXiv preprint arXiv:2303.07849* (2023).

**Questions:**

see Weaknesses

---

### Official Review · Reviewer_8Yx7 · 2023-10-30

**Soundness:** 2 fair
**Presentation:** 3 good
**Contribution:** 2 fair
**Rating:** 6
**Confidence:** 3

**Summary:**

USTEP (Unified Spatio-TEmporal Predictive Learning) is an innovative framework that addresses the integration of recurrent-based and recurrent-free methods, combining microtemporal and macro-temporal scales. This framework represents a significant advancement in the field of spatio-temporal predictive learning.

The aim of USTEP is to reconcile the strengths of recurrent-based methods, which excel in capturing short-term dependencies and microtemporal patterns, with the advantages of recurrent-free methods, which are effective in modeling long-term dependencies and macro-temporal trends. By integrating both scales, USTEP provides a unified approach that leverages the benefits of both methodologies.

**Strengths:**

1. Integration of Microtemporal and Macro-temporal Scales: USTEP effectively combines the strengths of recurrent-based methods and recurrent-free methods by integrating both microtemporal and macro-temporal scales. This integration allows the framework to capture short-term dependencies and microtemporal patterns, as well as long-term dependencies and macro-temporal trends, providing a comprehensive understanding of spatio-temporal data.

2. Improved Accuracy and Prediction Performance: By reconciling the two scales, USTEP enhances the accuracy and prediction performance of spatio-temporal models. It leverages the local spatial and temporal dependencies captured by recurrent-based models, while also considering the global patterns and trends captured by recurrent-free models. This comprehensive approach results in more accurate predictions and a better understanding of the underlying dynamics of the data.

3. Flexibility and Adaptability: USTEP is designed to be flexible and adaptable to various types of spatio-temporal data. It can incorporate different architectural components, such as RNNs, CNNs, transformers, or GNNs, depending on the characteristics of the data and the specific predictive task. This flexibility allows researchers and practitioners to tailor the framework to their specific needs and datasets.

**Weaknesses:**

1. Complexity and Computational Cost: Integrating both microtemporal and macro-temporal scales in the USTEP framework can result in increased complexity and computational cost. The combination of recurrent-based and recurrent-free models may require more computational resources and longer training times compared to using a single approach. This could potentially limit its practicality in real-time or resource-constrained applications.

2. Model Selection and Hyperparameter Tuning: USTEP incorporates multiple architectural components and mechanisms, which introduces additional complexity in model selection and hyperparameter tuning. Determining the optimal combination of models and hyperparameters can be challenging, requiring extensive experimentation and computational resources. This process may require domain expertise and thorough evaluation to achieve the best performance.

**Questions:**

has been proposed in weakness.

**Details Of Ethics Concerns:**

None.

---

### Official Review · Reviewer_nrq1 · 2023-10-31

**Soundness:** 3 good
**Presentation:** 2 fair
**Contribution:** 2 fair
**Rating:** 3
**Confidence:** 3

**Summary:**

The paper claims that there are two distinct approaches to temporal modeling in spatio-temporal learning: recurrent-based and recurrent-free methods, and that recurrent-based methods are better at capturing the cause and effect temporal relationship, but less efficient, and worse at capturing global information, while the recurrent-free methods are the opposite.
The paper proposed a novel view that unifies both these methods.

**Strengths:**

1. Extensive Analysis: The paper presents a comprehensive analysis of the proposed method. The use of multiple datasets and metrics provides a robust evaluation of the method’s performance. The complexity analysis via FLOPs is a valuable addition, offering insights into the computational efficiency of the method.

2. State-of-the-Art Performance: The method achieves state-of-the-art (SOTA) results, demonstrating its effectiveness compared to existing approaches.

3. Clarity of Presentation: The paper is well-written and easy to follow. The figures illustrated the concepts effectively.

**Weaknesses:**

The reccurance based/free duality presented here seems unprincipled.

In regards to the strengths and weaknesses of recurrent-based vs recurrent-free methods and the experiments:
Firstly, the second paragraph of the introduction section claimed that recurrent-based methods are better at capturing cause and effect temporal relationship, while the recurrent-free methods are better at capturing the global relationship.
Secondly, if the extended frame task favors the recurrent-based methods, then reduced frame task (instead of the equal frame task) should logicaly favors the recurrent-free method, but this is not supported by the empirical evidence.

The idea of a "unified view" seems to suggest that, under certain set of hyperparameters, USTEP should be a fully recurrent-based model, while under a different set, it would shouldbe a fully reccurent-free model.
However, it is not clear if such is actually the case since it is not discussed.

More importantly, my understanding is (despite the lack of explanation) that when delta-t = delta-T = 1, this should approximate a recurrent-based method, while when delta-t = delta-T = T, this should approximate a recurrent-free method.
Therefore the ablation study should study the combined effect of delta-t and delta-T, at both reduced and extended frames as well, to empirically test the earlier claim regarding the strengths and weaknesses of recurrent-based vs recurrent-free methods

Minor comments:
Figure 1, 2, and 3 should be combined and placed on the first page to act as a visual abstract.

Definition 3.3 says that macro-temporal scale is non-overlapping. But section 4.1 and figure 3 says that it is overlapping.

Adding a figure that capture the entire system architecture, including F(), and the cross segment modules would be helpful.

Appendix A should include the hyperparameter search space and the best hyperparameter found.

In Figure 7, |target-predict| seems to be completely black.

**Questions:**

1. Is it true that when delta-t = delta-T = 1, USTEP approximate a recurrent-based method, while when delta-t = delta-T = T, USTEP approximate a recurrent-free method?

2. The claims in the first paragraph of the experiments section regarding the how and why the different tasks capture a different aspect of model capabilities.

3. A more complete ablation as previously described.